# Radiation-Induced Overexpression of TGFβ and PODXL Contributes to Colorectal Cancer Cell Radioresistance through Enhanced Motility

**DOI:** 10.3390/cells10082087

**Published:** 2021-08-13

**Authors:** Hyunjung Lee, Joon-Seog Kong, Seung-Sook Lee, Areumnuri Kim

**Affiliations:** 1Laboratory of Radiation Exposure & Therapeutics, National Radiation Emergency Medical Center, Korea Institute of Radiological & Medical Science (KIRAMS), Seoul 01812, Korea; sadupia@kirams.re.kr (H.L.); balltta9@kirams.re.kr (J.-S.K.); sslee@kirams.re.kr (S.-S.L.); 2Department of Pathology, Korea Cancer Center Hospital, Korea Institute of Radiological & Medical Science, Seoul 01812, Korea

**Keywords:** colorectal cancer, PODXL, radioresistance, TGFβ

## Abstract

The primary cause of colorectal cancer (CRC) recurrence is increased distant metastasis after radiotherapy, so there is a need for targeted therapeutic approaches to reduce the metastatic-relapse risk. Dysregulation of the cell-surface glycoprotein podocalyxin-like protein (PODXL) plays an important role in promoting cancer-cell motility and is associated with poor prognoses for many malignancy types. We found that CRC cells exposed to radiation demonstrated increased TGFβ and PODXL expressions, resulting in increased migration and invasiveness due to increased extracellular matrix deposition. In addition, both TGFβ and PODXL were highly expressed in tissue samples from radiotherapy-treated CRC patients compared to those from patients without this treatment. However, it is unclear whether TGFβ and PODXL interactions are involved in cancer-progression resistance after radiation exposure in CRC. Here, using CRC cells, we showed that silencing PODXL blocked radiation-induced cell migration and invasiveness. Cell treatment with galunisertib (a TGFβ-pathway inhibitor) also led to reduced viability and migration, suggesting that its clinical use may enhance the cytotoxic effects of radiation and lead to the effective inhibition of CRC progression. Overall, the results demonstrate that downregulation of TGFβ and its-mediated PODXL may provide potential therapeutic targets for patients with radiotherapy-resistant CRC.

## 1. Introduction

Colorectal cancer (CRC) has the third-highest incidence and fatality rate worldwide, and while the surgical removal of CRC metastatic lesions continues to provide the best prognosis for these patients, it is insufficient to reduce recurrence [1]. CRC patient mortality is closely related to metastasis recurrence [2]. Despite the widespread use of chemo/radiotherapy in patients with inoperable CRC, these treatments have not changed the recurrence rate [3], and single-arm studies have shown that radiotherapy resulted in no significant overall survival improvement for patients with rectal cancer [4]. Furthermore, radiotherapy has been reported to actually promote distant rectal-cancer metastases [5]. Therefore, more in-depth studies of the molecular dynamics related to radiotherapy resistance are needed to develop potential treatment strategies to overcome metastasis in CRC patients.

Many of the cytokines secreted by tumor, immune, and normal cells are involved in promoting tumor-cell migration, invasion, and metastasis following radiotherapy. These cytokines and growth factors can enhance cell motility, thus affecting tumor-cell invasiveness and migration [6,7,8,9]. Among these, transforming growth factor β (TGFβ) plays a critical role as a powerful inducer of metastatic capacity in tumor cells during epithelial-mesenchymal transition (EMT) progression and as a regulator of tight junctions [10]. The upregulation of TGFβ levels has been detected in primary tumor tissue from CRC patients, and this upregulation has been correlated with both metastasis and poor prognosis [11,12]. In addition, this TGFβ signaling was mainly observed in the central regions of both CRC invasions and liver metastases [13]. Most studies have focused on TGFβ as an essential factor that participates in tumor cell-cycle regulation and tumor growth, so a more in-depth understanding of dysregulated TGFβ mechanisms responsible for tumor metastasis is needed to increase the satisfactory responses to treatment [14].

Podocalyxin-like protein (PODXL) is a member of the CD34 family of cell-surface glycoproteins with a cytoplasmic tail. PODXL is known to play a crucial role in maintaining adequate cellular infiltration because of its anti-adhesive properties; it can regulate cell adhesion by dissociating from actin, resulting in the loss of podocyte integrity [15,16,17]. PODXL has been reported to be overexpressed in a variety of malignancies, and strong cell-membrane expression has been associated with both tumor aggressiveness and poor prognoses in a variety of tumors, including CRC [18,19,20,21]. In addition, PODXL has been shown to act as an EMT mediator, to be involved in metastatic behavior, and to induce actin recruitment in several types of cancer cells [22,23,24]. Even though a correlation between PODXL expression and tumor prognostic value has been reported [24], details of an underlying mechanism remain unclear.

In the present study, we determined that IR-induced TGFβ and PODXL promoted CRC progression by regulating cell motility and invasiveness. In addition, treatment with galunisertib (a TGFβ-pathway inhibitor) dysregulated PODXL and repressed both CRC cell growth and motility. Our findings suggest that targeting the TGFβ-mediated PODXL may be a promising therapeutic approach for CRC treatment, particularly in radiotherapy-resistant patients.

## 2. Materials and Methods

### 2.1. Cell Culture, Irradiation, and Drug Treatment

Human colorectal cancer cell lines (HCT116, LoVo, SW620, SW480, DLD1, T84, HT29, and SW837) were obtained from the ATCC. Cells were cultured in RPMI medium containing 10% heat-inactivated fetal bovine serum (Gibco, Carlsbad, CA, USA), 100 units (U)/mL penicillin, and 100 U/mL streptomycin (Gibco) in a humidified atmosphere of 5% CO_2_ at 37 °C. SW837 cells were cultured under normal atmospheric conditions at 37 °C. Cells were irradiated with 137Cs γ-rays using a Gamma Cell-3000 irradiator (MDS Nordion International). Galunisertib (LY2157299) was obtained from Selleckchem.

### 2.2. Patient Samples

To compare PODXL tissue expression between non-tumor, non-metastatic tumor, and metastatic-tumor samples, human tissue microarray slides (containing 44 CRC tissue and non-tumor tissue pairs) were obtained from ISU ABXIS (Seoul, Korea). All slides were prepared using paraffin-embedded human tissue. Three patient-derived tumor samples and three normal samples were obtained from the Korea Institute of Radiological and Medical Sciences (Seoul, Korea) for Western blot analysis.

### 2.3. RNAi Knockdown and Plasmid Transfection

Lentiviral short hairpin RNAs (shRNA) targeting PODXL and control RNA were purchased from Sigma-Aldrich (St. Louis, MO, USA). Lentivirus was generated using 293T cells and the shRNA vector using Lipofectamine 2000 and Plus reagents (Thermo Fisher Scientific, Waltham, MA, USA). Viral particles were added to HCT116 and DLD1 cells and then incubated for 48 h. Puromycin (2 mg/mL) was added to the cultures to select transduced cells. The PODXL plasmid was obtained from OriGene (NM-001018111). SW480 cells were transiently transfected with either a control vector (pCMV6-XL3) or wild-type human PODXL using Lipofectamine 2000 and then maintained for 48 h. Cells overexpressing PODXL were selected using hygromycin B (Sigma-Aldrich).

### 2.4. Transwell Assay

To assess cell migration and invasiveness, irradiated cells were seeded into 24-well transwell chambers (Corning, NY, USA) with 8 μm pore polycarbonate membrane-coated inserts. For the invasion assay, cells (2 × 104) were seeded into the upper chambers coated with Matrigel (BD Biosciences, La Jolla, CA, USA). For the migration assay, irradiated cells were seeded into the upper chambers in serum-free medium, and 700 μL RPMI 1640 (supplemented with 10% fetal bovine serum) was added to the lower compartments at 37 °C for 24 h. Cells were fixed using 4% paraformaldehyde for 10 min, stained with 0.1% cresyl violet for 20 min, and then photographed using a bright-field microscope (Olympus, Shinjuku, Japan). The reported values represent (at least) triplicate experiments. Data values are mean ± standard deviation (SD).

### 2.5. Cell Viability and Clonogenic Assay

To investigate cell growth following radiation treatment, irradiated cells were seeded into 60-mm dishes (LoVo and SW480 cells, 4 × 10^3^; HCT116, and DLD1 cells, 3 × 10^3^) and maintained for 21 days in media. All culture media was renewed every three days. To measure cell survival following treatment with galunisertib (LY2157299) or radiation, irradiated cells were seeded in 60-mm dishes cultured for 3 days in media containing the indicated concentrations of LY2157299. Colonies were stained with 1% methyl blue in methanol, and colonies containing 50 cells or more were counted as clonogenic cells using a microscope (Leica). The survival-fraction values reported represent the means of six replicates from a minimum of three independent experiments. Cell viability was determined using a colorimetric assay based on tetrazolium salt cleavage (CCK-8 Assay Reagent; Abcam, Cambridge, UK). HCT116 cells (3 × 103/well) were seeded into 96-well plates and treated with either LY2157299 (10 μM) or radiation alone, or the two combined, for 24 h. After cell incubation with CCK-8 for 2 h, absorbance at 450 nm was measured using a SpectraMax M3 microplate reader (Thermo Fisher Scientific, Waltham, MA, USA). Relative cell numbers were determined using a comparison to control cells. Data values are mean ± SD from three independent experiments.

### 2.6. Flow Cytometry Analysis

For cell-cycle and apoptotic-cell assessments, HCT116 cells (4 × 10^5^) were treated with either 10 μM LY2157299 alone, radiation alone, or the two combined, for 24 h. To detect apoptotic cells, cells were stained with Annexin V-FITC and propidium iodide (PI; BD Biosciences, Franklin lakes, NJ, USA). Cellular DNA content was assessed using a FACSCanto II flow cytometer (BD Biosciences, San Jose, CA, USA) and the data were analyzed using FlowJo software (TreeStar). For analysis of mitochondria membrane potential, HCT116 cells (4 × 10^5^) were treated with either 10 μM LY2157299 alone, radiation alone, or the two combined, for 24 h. To analyze mitochondria membrane potential, cells were stained with JC-1 (Thermo Fisher Scientific) and assessed using a FACSCanto II flow cytometer (BD Biosciences).

### 2.7. Wound-Healing Assay

To observe migration ability, HCT116 and DLD1 cells (1 × 10^5^) were plated into 6-well plates, and the cell monolayer was manually scratched using a 200 μL pipette tip. Cells were maintained in growth media for 12 h under the transfection and ionizing radiation (IR) conditions indicated. Images were captured using a bright-field microscope (Leica, Wetzlar, Germany). Percent wound-closure was calculated according to the wound area at each time point compared to the initial (0 h) wound area.

### 2.8. Western Blot Analysis

Cells were lysed using NP40 lysis buffer (Pierce, Rockford, IL, USA) containing 1 mM sodium orthovanadate, pH 7.4, for 30 min and then centrifuged at 12,000 rpm for 30 min. Protein content of the cell lysates was determined using the Bradford assay (Pierce). Equal protein amounts (50–100 μg) were separated using SDS-PAGE and then transferred to polyvinylidene difluoride membranes. Membranes were blocked for 1 h in 5% nonfat dry milk in Tris-buffered saline with Tween 20 (Thermo Fisher Scientific) and incubated overnight with primary antibodies against the following proteins: PODXL (Abcam), TGFβ (Abcam), ZO1, vimentin, p-FAK, cleaved caspase 3, cleaved PARP, Bcl-2, Bcl-xL, tBid, FAK (Cell Signaling Technology, Danvers, MA, USA), and β-actin (Santa Cruz Biotechnology, Santa Cruz, CA, USA). Membranes were incubated with secondary antibodies for 2 h. Immunopositive bands were visualized using enhanced chemiluminescence detection reagent (GE Healthcare, Uppsala, Sweden).

### 2.9. Immunofluorescence (IF) Staining

Cells were seeded onto slides and incubated for 24 h. After washing three times using phosphate-buffered saline, cells were fixed using 4% paraformaldehyde for 10 min at room temperature. After incubation with H_2_O_2_ for 30 min to block peroxidase activity, cells were incubated overnight at 4 °C with either anti-PODXL and anti-TGFβ, or anti-p-FAK and anti-ZO1. IF staining was visualized using incubations with Alexa Fluor 488- or Alexa Fluor 594-conjugated secondary antibody (Invitrogen) for 2 h at room temperature. Nuclei were counterstained with DAPI (0.2 μg/mL; Sigma-Aldrich) for 10 min at room temperature. Images were captured using a microscope (Leica).

### 2.10. Immunohistochemistry (IHC) Staining

Tissue samples were fixed in formalin and embedded in paraffin. Sections (4 μm) were prepared, dried, and deparaffinized. To block endogenous peroxidase activity, slides were incubated for 30 min in methanol containing 0.3% H_2_O_2_. Antigen retrieval was performed using citrate buffer in a steamer for 30 min. Sections were incubated with a primary antibody against PODXL, followed by detection using a LabVision horseradish peroxidase polymer detection system (Thermo Fisher Scientific). PODXL staining was assessed by the percentage of IHC-positive cells (1 = 0–25%; 2 = 26–50%; 3 = 51–75%; 4 = 76–100%). IHC staining intensity for PODXL was scored between 0 and 3 (0 = none; 1 = weak; 2 = moderate; 3 = strong). The level of PODXL staining in each sample was calculated as staining intensity × percentage of positive cells. Images of immunostained slides were captured using a light microscope (Olympus).

### 2.11. Bioinformatics Analysis

To determine the role of PODXL in CRC, we analyzed the Human Protein Atlas (HPA) database (https://www.proteinatlas.org/, Accessed 20 January 2021), which contains data from The Cancer Genome Atlas (TCGA) project database (https://tcga-data.nci.nih.gov/tcga/, Accessed 20 January 2021) and includes both gene-expression data and follow-up information. PODXL-expression data from 597 CRC patients were included in this study. Patients were divided into two groups based on their PODXL expression levels. Patients with an expression level greater than the median were classified as the high-expression group, and those with median or lower expression levels were classified as the low-expression group.

### 2.12. Spheroid Invasion Assays

Cells were seeded in 96-well plates coated with matrigel in serum-free RPMI1640 and cultured on a shaker at 100 rpm for 48 h. After single spheroids formed, cells were treated with IR or LY2157299 alone or in combination with IR (5Gy). Treated spheroids were placed in the center of 96-well plates and were cultured at 37 °C for 72 h in serum-free RPMI1640. Invasion was monitored by taking pictures under a light microscope (Leica) immediately after implantation and after 72 h. Representative images were compiled without further adjustments. Depicted spheroids are representative of three independent experiments.

### 2.13. Matrix Degradation Assay

HCT116 cells (1 × 10^6^) were assessed for matrix degradation using the QCM ECMatrix Cell Invasion Assay Kit (Millipore) following the manufacturer’s protocol. Images were captured using a microscope (Leica), and degraded areas were measured using ImageJ software. Area measurements were then normalized according to the number of cells.

### 2.14. Statistical Analysis

Each experiment was performed in triplicate, and the results were expressed as mean ± SD. Significant differences between groups were determined using Student’s *t*-tests as appropriate using GraphPad Pro software (GraphPad, San Diego, CA, USA). For these comparisons, a *p* value < 0.05 was considered to be statistically significant.

## 3. Results

### 3.1. Ionizing Radiation (IR) Exposure Promoted CRC Cell Migration

To assess whether IR might affect CRC cell motility differently based on radiosensitivity, we selected two IR-resistant cell lines (HCT116 and DLD1) and two IR-sensitive cell lines (LoVo and SW480) by clonogenic assay (Appendix A). The HCT116 and DLD1 cells were exposed to IR, and then the infiltration rates were measured using a migration assay. As expected, IR did not affect the migration of IR-resistant cells, whereas it greatly reduced the migration of IR-sensitive cells (Figure 1a). IR-exposed HCT116 and DLD1 cells showed spindle-shaped and elongated morphologies, which are known to be typical of EMT morphological phenotypes (Figure 1b). IR exposure also decreased the expression of ZO1 (a tight-junction protein) and E-cadherin (an epithelial marker), while increasing vimentin (a mesenchymal marker) and Snail in both HCT116 and DLD1 cells (Figure 1c). Overall, IR elevated CRC migration, suggesting that enhanced TGFβ might be involved in this movement.

### 3.2. IR-Enhanced Migratory Potential through Upregulation of TGFβ and PODXL

To determine whether IR altered TGFβ and PODXL protein levels, we assessed these proteins in tissues from both radiotherapy-treated CRC patients and those without this treatment. At identical tissue sites, the levels of TGFβ and PODXL were both elevated, whereas E-cadherin protein levels decreased in tissues from IR-exposed patients compared to that in tissues from non-exposed patients (Figure 2a). Next, to determine whether IR exposure also changed the levels of TGFβ and PODXL expression in vitro, CRC cells were exposed to IR. The expressions of TGFβ and PODXL were increased after IR exposure in HCT116 and DLD1 cells, but they were not significantly changed in IR-sensitive cells (Figure 2b). Consistent with our Western blot data, immunofluorescence staining also revealed that IR exposure increased the co-localized expressions of PODXL and TGFβ in HCT116 cells (Figure 2c). We next stained for actin and phosphorylated focal adhesion kinase (p-FAK) to assess IR-induced lamellipodia formation, indicating migratory cells. As shown in Figure 2d, IR-exposed HCT116 cells showed more lamellipodia structures compared to that of control cells. p-FAK staining also showed increased fluorescence intensity (>50% increase) after IR exposure (Figure 2e). These results demonstrate that IR exposure stimulated migration ability through the elevated expressions of both PODXL and TGFβ in IR-resistant cells.

### 3.3. Correlation between Upregulated PODXL and Poor Prognoses in CRC Patients

Upregulated PODXL has been reported to be associated with tumor progression via increased cell migration and invasion [25]. Therefore, to validate the effect of PODXL expression on CRC metastasis, PODXL protein levels were evaluated using IHC in 44 CRC tumor-tissue samples and in 20 normal-tissue samples. We found that PODXL protein levels were high in the metastatic-tissue samples compared to the non-metastatic samples (Figure 3a). As expected, PODXL protein was rarely expressed in normal tissues, and in addition, both ZO1 and E-cadherin levels were decreased in the CRC tissue compared to those in the non-tumor tissue (Figure 3b). We also assessed patient survival rates according to PODXL expression using TCGA data. CRC patients with high PODXL expression showed distinctly worse outcomes than those with low PODXL expression (Figure 3c). Collectively, these data suggest that PODXL may play an important role in CRC cell movement to promote metastasis.

### 3.4. PODXL Knockdown Decreased Cell Motility through the Regulation of EMT-Related Gene Expression

Given the association between PODXL expression and the poor prognoses of patients with CRC, we examined the mechanism by which PODXL increases cancer progression in CRC cells. We therefore established stable PODXL-silencing in both HCT116 and DLD1 cell lines using shRNA. Figure 4a shows that depletion of PODXL substantially reduced EMT-related gene expression, including the inhibition of both p-FAK and vimentin expression. To assess the functional impact of PODXL expression in cancer-cell migration, PODXL knockdown in both cell types reduced wound healing at the indicated time points (Figure 4b) when compared to that of the control cells. In addition, both the migratory and invasive potentials were significantly inhibited (>50% reduction) in PODXL-knockdown (shPODXL) HCT116 and DLD1 cells compared to those of control (shCont) cells (Figure 4c). However, PODXL depletion did not significantly suppress CRC cell growth (Appendix A). Together, these data demonstrate that PODXL depletion blocked cell migration and invasiveness by regulating EMT-related gene expression in these CRC cells.

### 3.5. PODXL Deletion Inhibited Invasiveness after IR Exposure

We next assessed whether PODXL knockdown could affect CRC motility in IR-exposed cells. Although IR exposure increased cell migratory ability, with a significant reduction (~40%) in shPODXL-cell motility compared to that of shCont cells (Figure 5a). To determine whether shPODXL cells could degrade the extracellular matrix (ECM), we used an invadopodia-based matrix-degradation assay (loss of FITC-conjugated gelatin). Figure 5b shows the IR-enhanced matrix degradation of shPODXL cells (dark areas in the FITC-tagged gelatin matrix) compared to that of untreated shCont cells. In addition, blocking PODXL expression had no effect on matrix degradation with or without IR exposure, whereas IR exposure alone enhanced shCont-cell ECM degradation. Consistent with these results, mesenchymal marker proteins (e.g., vimentin, p-FAK, Snail, and matrix metallopeptidase 2) were reduced and IR exposure did not increase these proteins in shPODXL cells (Figure 5c). Similarly, immunofluorescence staining revealed that both p-FAK and TGFβ expressions were suppressed by PODXL depletion in IR-exposed cells (Figure 5d). In shPODXL cells, ZO1 was localized at intercellular borders in a belt-like manner, encircling the cells after IR exposure. In contrast, ZO1 staining was punctate, with some loss from membranes, in IR-exposed shCont cells (Figure 5e). These data indicate that PODXL downregulation blocked IR-induced migration and invasiveness in CRC cells through mesenchymal alterations.

### 3.6. PODXL Overexpression Enhanced Cell Motility in IR-Sensitive CRC Cells

To confirm PODXL function as a key mediator of radiation-induced migration, we transfected radiosensitive SW480 cells with a PODXL-overexpressing vector. As shown in Figure 6a, PODXL upregulation increased both the migratory and invasive capacities of SW480 cells compared to those of control cells, and its overexpression significantly blocked (>50% reduction) cell motility induced by IR exposure. In addition, these SW480-PODXL cells showed p-FAK upregulation and ZO1 inhibition compared to those of SW480 cells after IR alone (Figure 6b). Immunofluorescence staining also showed that PODXL overexpression significantly decreased ZO1 belt-like expression at intercellular boundaries (Figure 6c). These observations indicate that PODXL upregulation affected resistance to IR-induced cell motility via increased FAK expression and reduced ZO1 expression.

### 3.7. Galunisertib Enhanced IR-Antitumor Activity in CRC Cells

To determine whether TGFβ-pathway inhibition by galunisertib also suppressed PODXL and associated proteins, we combined IR and galunisertib treatments in HCT116 cells (Figure 7a). Galunisertib alone markedly diminished PODXL, p-FAK, and vimentin levels but increased the level of E-cadherin. However, IR treatment combined with galunisertib reduced PODXL levels even further compared to that of galunisertib treatment alone. Moreover, galunisertib enhanced both IR-induced cell growth and apoptosis compared to that of either treatment alone (Figure 7b,c and Appendix A) through reduced p-AKT, Bcl-2, and Bcl-xL (anti-apoptotic proteins) levels but increased tBid, cleaved caspase-3 and cleaved PARP levels (Appendix A). Additionally, combined treatment of galunisertib and IR enhanced mitochondrial dysfunction (Appendix A). Migratory ability was also significantly repressed (>50% reduction) by this combined treatment compared to that of either treatment alone (>30% reduction) (Figure 7d). Consistent with in the vitro results, the antitumor activity of galunisertib was confirmed using a spheroid invasion assay (Figure 7e). These data clearly indicate that galunisertib inhibition of the TGFβ pathway suppressed IR-induced motility through PODXL downregulation. In addition, this TGFβ-pathway inhibitor enhanced the cytotoxic effects of IR exposure in these CRC cells.

## 4. Discussion

Here, we demonstrated that inhibition of the TGFβ pathway enhanced the cytotoxic effects of IR and led to suppressed CRC progression. Our experiments showed that IR exposure increased TGFβ and accelerated both cell migration and invasion through PODXL overexpression in CRC cells. We also found that both TGFβ and PODXL were aberrantly expressed in CRC patients who received radiotherapy. Moreover, a TGFβ-pathway inhibitor, galunisertib, suppressed PODXL activation and reduced both CRC cell viability and their ability to migrate. Thus, our results indicate that suppression of the TGFβ pathway may inhibit PODXL, implying a new potential strategy for blocking metastasis in CRC patients.

Approximately half of all patients with solid cancers receive radiotherapy, but some cancers acquire radioresistance [26] that covers migration and invasiveness, which are essential components of cancer progression and metastasis [27]. Therefore, it is necessary to find novel therapeutic approaches for improving the prognoses of CRC patients who relapse after radiotherapy. Many studies have attempted to overcome radioresistance by developing targeted agents to prevent radiation-induced metastasis. One study reported that radiation combined with an inhibitor of K-Ras/c-Raf/p38 signaling eliminated the metastatic potential of cervical cancers cells [28]. In addition, IR combined with treatment using a mitogen-activated protein kinase 1 inhibitor showed reduced breast-cancer cell migration [29]. For CRC, the pharmacological inhibition of phosphoinositide 3-kinase/AKT was shown to markedly enhance the cytotoxic effects of IR both in vitro and in vivo [30]. Even though many studies have reported mechanisms underlying radioresistance-enhanced cell invasiveness and the prevention of IR-induced cancer-cell motility, these mechanisms remain poorly characterized in CRC.

It is well known that TGFβ primarily functions as a tumor suppressor in premalignant cells, but in cancer cells, it acts as a metastasis promoter that regulates downstream effector signaling [31,32]. TGFβ promotes tumor progression through the stimulation of angiogenesis and metastasis as part of the cancer-cell radiation response, leading to poor patient prognoses [33]. Accordingly, previous studies have shown that TGFβ suppression increased the radiosensitivity of glioblastoma cells and delayed breast-cancer growth [34,35]. Consistent with these reports, our data demonstrated that IR enhanced TGFβ expression and increased both cell motility and EMT in CRC cells (Figure 1 and Figure 2). A previous study showed that treating with an inhibitor of type I TGFβ receptors reduced IR-induced cell migration via blocked Smad signaling in vitro. In addition, targeting TGFβR I/II with LY2109761 alone or in combination with radiation also reduced the ability of glioblastoma to migrate [36]. These findings led us to investigate the possibility that inhibiting TGFβ-related signaling might prevent the increased EMT ability induced by IR.

PODXL overexpression has been shown to be associated with metastasis in a variety of cancer-tissue types and cancer-cell lines [15,16,19]. In particular, the membranous expression of PODXL has been reported to be a biomarker for improved treatment stratification of patients with periampullary adenocarcinoma [37]. Furthermore, other research has shown a positive correlation between dysregulated PODXL expression and poor prognoses for metastasis, likely associated with its link to the epidermal growth factor receptor axis [38]. We also found that membranous PODXL was highly overexpressed in metastatic samples from CRC patients compared to that from non-metastatic samples, and that higher PODXL expression was associated with shorter overall CRC-patient survival based on TCGA data (Figure 3).

While a recent study by Meng et al. reported a marked increase in PODXL expression during the TGFβ-induced EMT process seen in lung cancer [39], the biological functions and cellular responses to radiation-induced PODXL remain unclear. In this study, we first demonstrated that dysregulated TGFβ signaling induced by IR exposure contributed to elevated PODXL expression, thereby increasing CRC cell motility (Figure 2). We also found that silencing PODXL led to reduced cell motility via inhibition of p-FAK in IR-exposed cells (Figure 3 and Figure 4). Notably, CRC tissues from patients who received radiotherapy showed high expressions of both TGFβ and PODXL compared to those from samples from patients who did not receive radiotherapy (Figure 2). Our findings therefore demonstrate that PODXL-mediated TGFβ signaling may promote CRC metastasis and provide evidence for an attractive therapeutic target for overcoming radioresistance. Other studies have reported that PODXL inhibition greatly increased cell death in breast cancer and in lung cancer [19,25], but we found that PODXL knockdown did not significantly change cell viability with IR or not, suggesting that PODXL is markedly associated with cell movement rather than cell growth in CRC (Appendix A).

Recently, clinical studies have provided evidence that targeting the TGFβ pathway is a useful treatment for cancer [40]. Galunisertib, a small-molecule inhibitor of TGFβ receptor I, has been reported to stop activation of this canonical pathway [40] and to block all three TGFβ ligands [40]. Galunisertib inhibition has been demonstrated in TGFβ-dependent tumor cells both in vitro and in vivo, and it has been shown to suppress tumor growth in mouse models. Furthermore, galunisertib is currently under clinical trials (NCT02423343; NCT02734160) development in combination with checkpoint inhibitors in patients with a variety of cancers. Here, we demonstrated that galunisertib treatment significantly downregulated PODXL expression and contributed to reduced cell migration capacity of CRC. In addition, the combination of a PODXL blockade using galunisertib and IR exposure enhanced the inhibition of cell viability and migration, suggesting that the IR-induced increase in TGFβ requires inhibition to overcome radioresistance (Figure 7).

In summary, we demonstrated that PODXL levels were significantly increased in CRC patients treated with radiotherapy, leading to an enhanced capacity for metastasis. Inhibitory treatment of the TGFβ pathway reduced IR-induced PODXL expression, which then decreased cell viability and migration. The combined treatments of galunisertib and IR exposure enhanced not only movement blocking but also CRC cytotoxicity. Collectivity, the inhibition of TGFβ may enhance the attenuation of PODXL expression, improving outcomes for CRC patients who relapse after radiotherapy.

## Figures and Tables

**Figure 1 cells-10-02087-f001:**
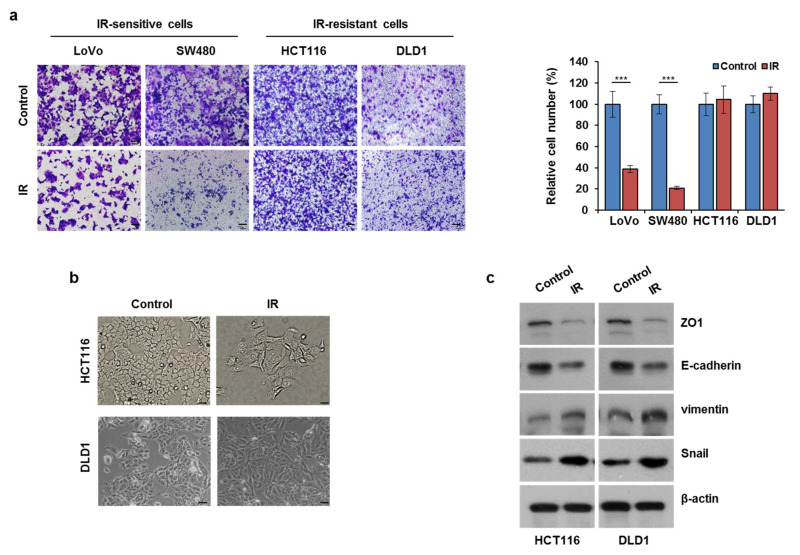
IR exposure enhanced EMT progression and migratory ability in CRC cells. (**a**) LoVo, SW480, HCT116, and DLD1 cells were exposed to IR at 5 Gy. After 24 h, cells were evaluated for migration ability using a transwell assay. The graph indicates the number of migrated cells. Scale bar, 100 μm. *** *p* < 0.005. (**b**) HCT116 and DLD1 cells were cultured with or without IR exposure (5 Gy) for 24 h. Cellular morphology was assessed using a microscope (magnification 20×). (**c**) HCT116 and DLD1 cells were exposed to IR at 5 Gy. Whole-cell lysates were subjected to Western blotting using antibodies against ZO1, E-cadherin, vimentin, and Snail. The β-actin signal was used as a loading control.

**Figure 2 cells-10-02087-f002:**
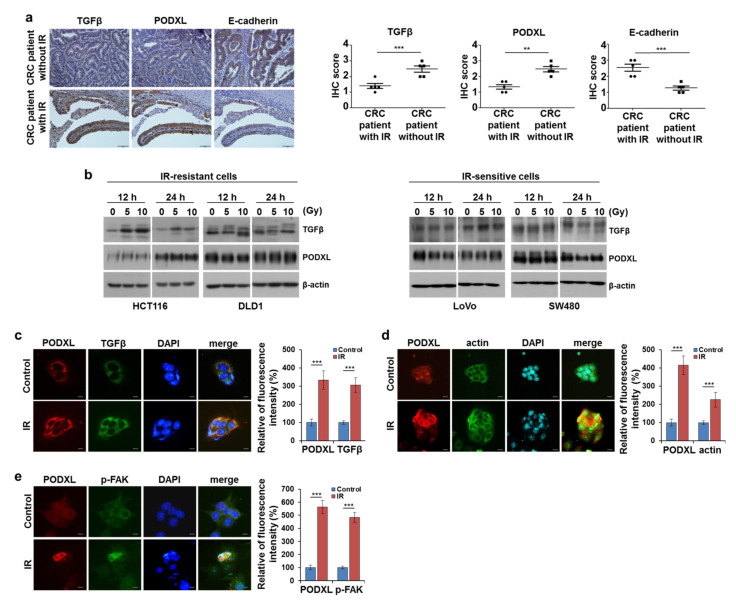
IR increased TGFβ and PODXL expressions in CRC cells. (**a**) PODXL, TGFβ, and E-cadherin expressions were stained (IHC) in CRC tissues from CRC patients with/without radiotherapy (RT). Statistical analyses of TGFβ, PODXL, and E-cadherin assessment scores were then determined for RT +/− CRC patients. Scale bar, 200 μm. (**b**) Cell lysates from the indicated cells were assessed for TGFβ and PODXL levels by Western blot following the indicated CRC-cell IR exposures (5 or 10 Gy) at 12 and 24 h. The β-actin signal was used as a loading control. HCT116 cells were IR-exposed (5 Gy) after 24 h. In panel (**c**), PODXL is red and TGFβ is green; in panel (**d**), PODXL is red and actin is green; and in panel (**e**), PODXL is red and p-FAK is green. DAPI was used for nuclear staining. Scale bar, 50 μm. The graph indicates fluorescence intensity from three independent experiments ± SD. ** *p* < 0.001, *** *p* < 0.005.

**Figure 3 cells-10-02087-f003:**
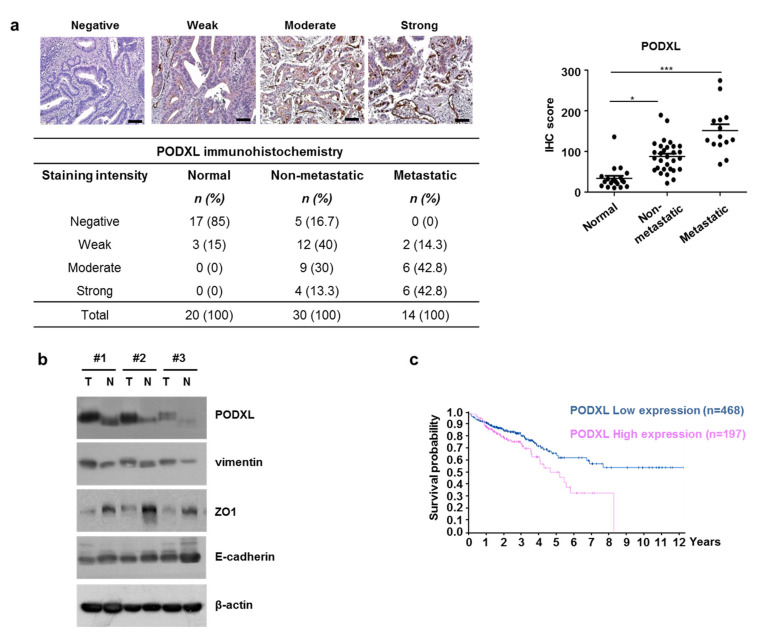
PODXL was upregulated in metastatic tumor tissue from CRC patients. (**a**) PODXL staining (IHC) was performed on tissues from CRC patients. Representative images are shown for negative, weak, moderate, and strong tissue staining from patients with stage I, II, III, and IV cancers, respectively. Statistical analyses of PODXL scores were determined for normal tissue, CRC tissue without metastasis, and CRC tissue with metastasis. Scale bar, 200 μm. (**b**) PODXL, vimentin, ZO1, and E-cadherin protein levels were assessed in paired clinical specimens from three CRC patients using Western blotting. N; Normal, T; Tumor. The β-actin signal was used as a loading control. (**c**) Assessment of survival probability in CRC patients according to PODXL expression levels using the Kaplan–Meier method and online resources (TCGA database). * *p* < 0.05, and *** *p* < 0.005.

**Figure 4 cells-10-02087-f004:**
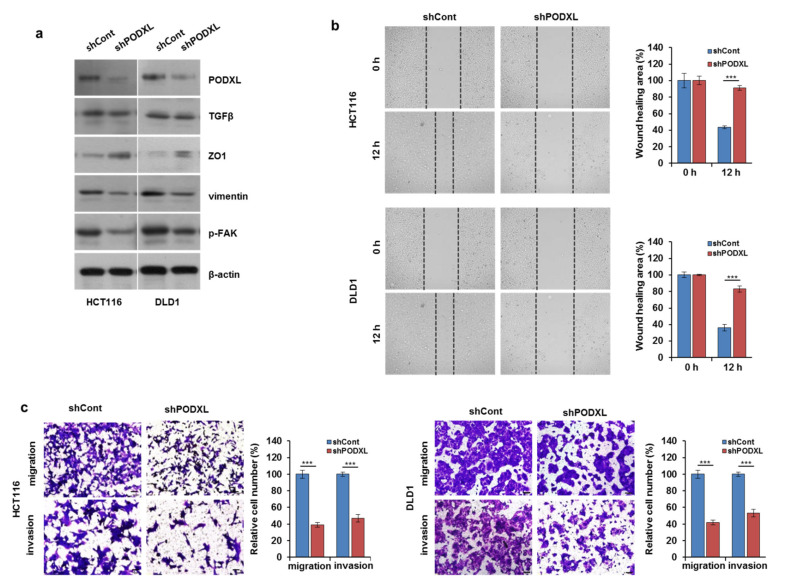
PODXL silencing attenuated both CRC migration and invasiveness. (**a**) HCT116 and DLD1 cells were transfected with lentiviral vector bearing either control shRNA (shCont) or shRNA against PODXL (shPODXL). Protein levels were then assessed using Western blotting. The β-actin signal served as a loading control. (**b**) Wound-healing ability was evaluated at 0 h and 12 h in shCont- or shPODXL-transfected HCT116 and DLD1 cells. *** *p* < 0.005. (**c**) The migration and invasion capabilities of shCont- or shPODXL-transfected HCT116 and DLD1 cells were investigated using 0.1% crystal violet staining. The graphs show relative cell numbers. Scale bar, 100 μm. *** *p* < 0.005. The data were derived from three randomly chosen fields from each of three independent experiments.

**Figure 5 cells-10-02087-f005:**
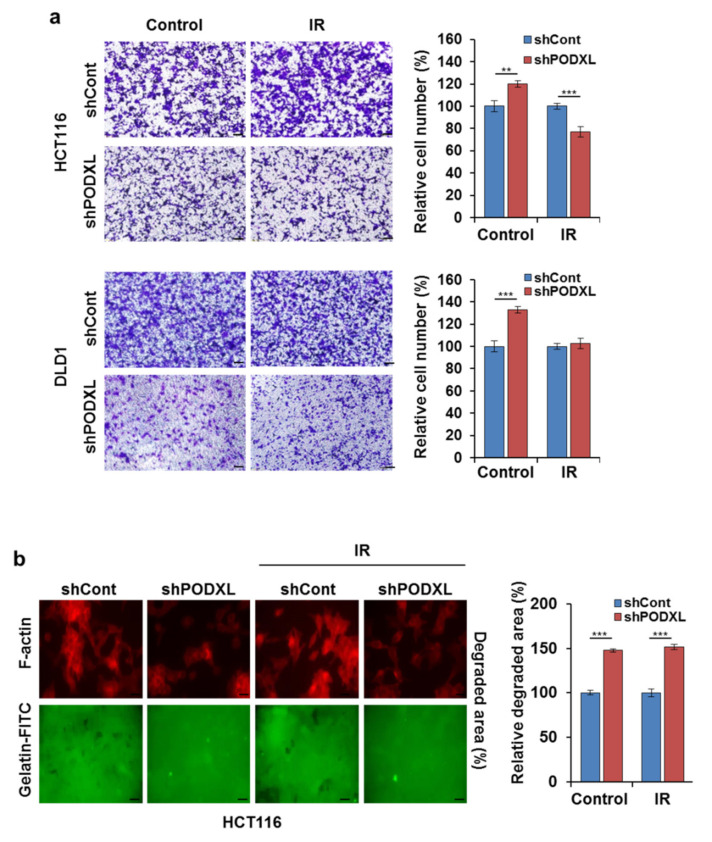
PODXL depletion inhibited invadopodia in CRC cells. (**a**) shCont and shPODXL cells were exposed to IR (5 Gy) and both migration and invasiveness were determined after 24 h using a transwell assay. The graphs show relative cell numbers. Scale bar, 100 μm. ** *p* < 0.001, and *** *p* < 0.005. (**b**) shCont- or shPODXL-transfected HCT116 cells were placed on FITC-coupled gelatin-coated coverslips and incubated for 48 h, either IR-untreated or IR-treated. Gelatin (green) degradation was visualized in areas devoid of FITC staining by fluorescent microscopy at a magnification of 20×. *** *p* < 0.005. (**c**) shCont- or shPODXL-transfected HCT116 and DLD1 cells were exposed to IR (5 Gy) and incubated for 24 h. Western blots were used to assess the expression patterns of PODXL, TGFβ, p-FAK, total FAK, ZO1, Snail, vimentin, and MMP2. The β-actin signal served as a loading control. shCont- or shPODXL-transfected HCT116 cells were IR-exposed (5Gy) or not, and then incubated for 24 h. In panel (**d**), cells were stained for p-FAK (red) and PODXL (green); in panel (**e**), the staining was for TGFβ (red) and ZO1 (green). DAPI was used for nuclear staining. The graph indicates fluorescence intensity from three independent experiments ± SD. * *p* < 0.05, ** *p* < 0.001, and *** *p* < 0.005.

**Figure 6 cells-10-02087-f006:**
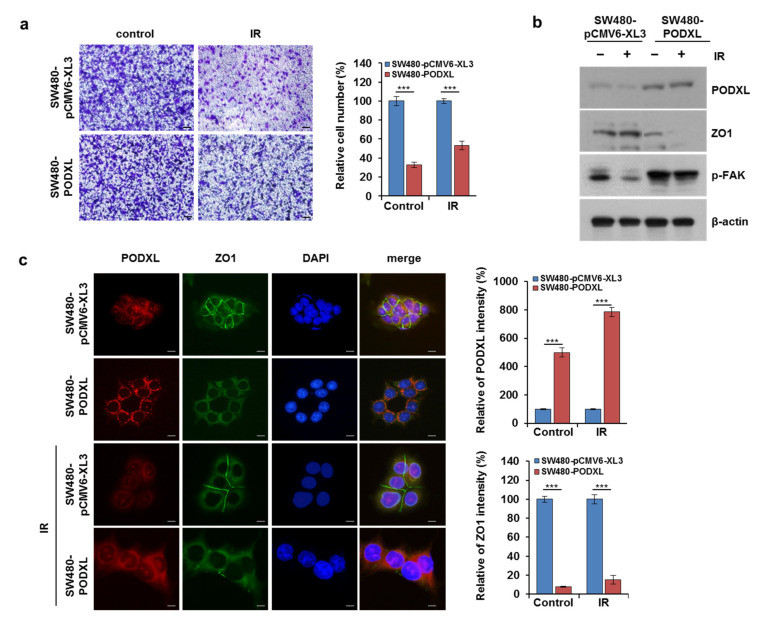
PODXL overexpression accelerated EMT progression and metastatic ability in CRC cells. (**a**) SW480 cells were transfected with control vector (pCMV6-XL3) or PODXL and then exposed to IR (5 Gy). Twenty-four hours after transfection, cell migration and invasiveness were determined using a transwell assay. The graph indicates relative cell numbers from three independent experiments. Scale bar, 100 μm. and *** *p* < 0.005. (**b**) pCMV6-XL3- or PODXL-expressing SW480 cells were assessed by Western blot for PODXL, ZO1, and p-FAK expression. The β-actin signal served as loading control. *** *p* < 0.005. (**c**) IF staining of PODXL (red) and ZO1 (green) in pCMV6-XL3- or PODXL-expressing SW480 cells. Nuclei were counterstained with DAPI. Scale bar, 100 μm. The graph shows fluorescence intensity ± SD. *** *p* < 0.005.

**Figure 7 cells-10-02087-f007:**
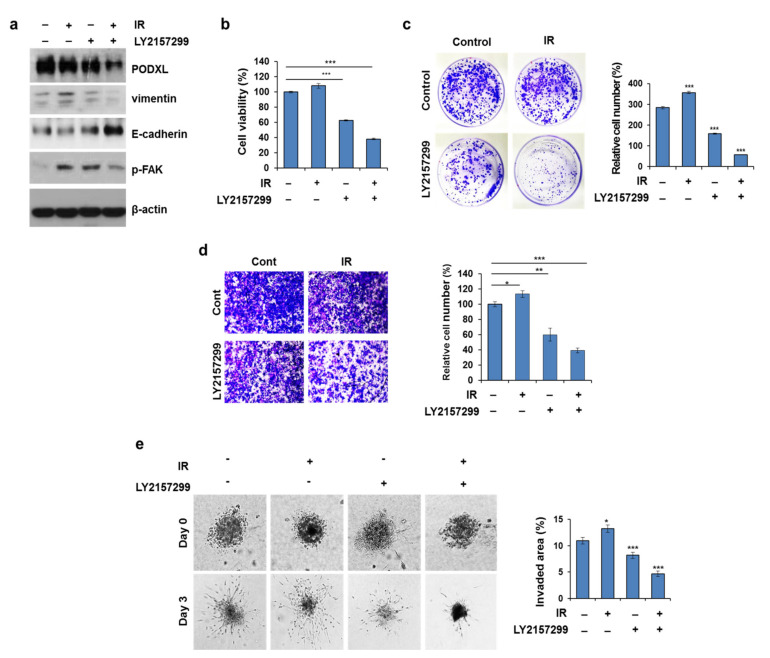
Combined galunisertib and IR treatment promoted antitumor activity in CRC cells. HCT116 cells were treated with galunisertib (10 µmol/L) alone, IR (5 Gy) alone, or combined for 24 h. (**a**) Western blot analysis was used to assess PODXL, vimentin, E-cadherin, and p-FAK expressions (**b**) HCT116 cells were treated with galunisertib (10 μM) alone, IR (5 Gy) alone, or combined for 24 h. Twenty-four hours after treatment, cell viability was estimated using a CCK-8 assay. The graph indicates cell viability from three independent experiments ± SD. *** *p* < 0.005. (**c**) C HCT116 cells were treated with galunisertib (10 µmol/L) alone, IR (5 Gy) alone, or combined for 21 days. The colony number was calculated from three replicate plates of three independent experiments; bars indicate SD. Representative images from 21 days post-plating are shown. (**d**) Cell migration was assessed using a transwell assay. The data shown are representative of three independent experiments. *** *p* < 0.005, ** *p* < 0.01, and * *p* < 0.05. (**e**) Representative images of the dissemination of HCT116 spheroids over a 3-day culture, in which cells were treated with IR, galunisertib, or combined treatment from day 0. Scale bar, 200 μm. The data shown are representative of three independent experiments. *** *p* < 0.005, ** *p* < 0.01, and * *p* < 0.05.

## Data Availability

Not applicable.

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
