# Peer review of "Radiation-Induced Overexpression of TGFβ and PODXL Contributes to Colorectal Cancer Cell Radioresistance through Enhanced Motility"

_cells, 2021, doi:10.3390/cells10082087_

Round 1

Reviewer 1 Report

The current article," Radiation-induced overexpression of TGFβ and PODXL contrib utes to colorectal cancer cell radioresistance through enhanced motility, " is interesting. Notably, Podocalyxin-like protein (PODXL) plays a role in promoting cancer-cell motility and is associated with poor prognoses for many malignancy types.

Here authors reported that colorectal cancer cells, when exposed to radiation, showed increased TGFβ and PODXL expressions resulting in enhanced migration and invasiveness.  In addition, both TGFβ and PODXL were highly expressed in human tissue samples obtained from radiotherapy-treated CRC patients.

By using CRC cells, they have shown that silencing PODXL blocked radiation-induced cell migration and invasiveness. Moreover, cancer Cells treated with galunisertib also led to reduced viability and migration. Overall, these results showed that downregulation of TGFβ and its-mediated PODXL may provide potential therapeutic targets for patients with radiotherapy-resistant colorectal cancer cells. Altogether paper is well written, and the results very well support the conclusion. 

In vivo studies: Tumor challenge and treatment experiments must be performed to assess the antitumor activity of PODXL and galunisertib in a colorectal cancer mouse model.  
Why Galunisertib enhanced both IR-induced cell growth and apoptosis compared to either treatment alone? What could be a possible mechanism for inducing cell death? Apoptosis-related molecules Bcl-2 and Caspase-3 or cytochrome release can be measured after treatment with galunisertib.

There is no direct evidence or relationship between PODXL expression and EMT induced by TGF‐β in the article.  Showing PODXL and EMT markers when CRC cells are treated with TGF‐β or with galunisertib is essential. 

Reviewer 2 Report

The authors investigated the function of TGFbeta and PODXL on colorectal cancer radio resistance. Overall the authors utilized different methods to support their conclusion. The over expression of TGFbeta and PODXL is confirmed in both CRC cells and patient tissue samples. Though previous studies have reported the importance of those two targets, this manuscript emphasized the importance of them on radio resistance especially in colorectal cancer cell. This improves our current limit understanding in radio resistance and brings insights in this field. The manuscript is well-organized and suggested to be accepted.

Reviewer 3 Report

Radiation-induced overexpression of TGFβ and PODXL contributes to colorectal cancer cell radioresistance through enhanced motility by Hyunjung Lee, et al shows relevant findings concerning to radioresistance and invasiveness colorectal cancer. Authors obtained relevant results through a righteous experimental design to find convincing conclusions.

However, authors didn´t show the response to radiotherapy of the cell lines used in the study.

Authors mentioned that LoVo and SW480 cells are radiosensitive cells while HCT116 and DLD1 cells are radioresistant, however they didn´t demonstrate these phenotypes by clonogenic assays after irradiation. Authors must demonstrate that these cells are resistant or sensitive to radiation. It´s important proved the radioresistance of the cells in order to conclude that activation of TGFb pathway in radioresistant cells might control the invasiveness phenotype in response to radiotherapy.

Round 2

Reviewer 1 Report

I have no more comment. 

Reviewer 3 Report

questions were answered